# Comparison of post-COVID symptoms in patients with different severity profiles of the acute disease visited at a rehabilitation unit

Jean Claude Perrot[1], Macarena Segura[1,2], Marta Beranuy[1], Ignasi Gich[3], Mª Josepa Nadal[1], Alberto Pintor[1], Jimena Terra[1], Eliot Ramirez[1], Luis Daniel Paz[1,2], Helena Bascuñana[1,2,3,4], Vicente Plaza[1,2,3,4], Mª Rosa Güell-Rous🄳[2]*

1 Physical Medicine and Rehabilitation Department, Hospital de la Santa Creu i Sant Pau, Barcelona, Catalonia, Spain, 2 Pneumology Department, Hospital de la Santa Creu i Sant Pau, Barcelona, Catalonia, Spain, 3 CIBER Epidemiology and Public Health (CIBERESP), Catalonia, Spain, 4 Sant Pau Biomedical Research Institute (IIB Sant Pau), Barcelona, Catalonia, Spain

* mguellr@santpau.cat

## Abstract

### Background and aim

Studies in the literature suggest the severity of COVID-19 may impact on post-COVID sequelae. We retrospectively compared the different patterns of symptoms in relation to the severity of acute COVID-19 in patients visited at our post-COVID rehabilitation unit.

### Methods

We compared respiratory, muscular, cognitive, emotional, and health-related-quality-of-life (HRQoL) measures in three groups of post-COVID patients: those who had not required hospitalization for the acute disease, those who had been admitted to a general hospital ward, and those who had been admitted to the ICU. The main inclusion criteria were persistent dyspnoea (mMRC $\geq$2) and/or clinical frailty (scale value $\geq$3).

### Results

We analyzed data from 178 post-COVID patients (91 admitted to the ICU, 60 to the ward, and 27 who had not required admission) at first visit to our post-COVID rehabilitation unit. Most patients (85.4%) had at least one comorbidity. There were more males in all groups (58.1%). ICU patients were older (p<0.001). The most frequent symptoms in all groups were fatigue (78.2%) and dyspnea (75.4%). Muscle strength and effort capacity were lower in the ICU group (p<0.001). The SF36 mental component and level of anxiety were worse in patients not admitted to the ICU (p<0.001). No differences were found between groups regarding respiratory pressure but 30 of 57 patients with a decrease in maximum inspiratory pressure had not required mechanical ventilation.

**Data Availability Statement:** All relevant data are within the article and its Supporting information files.

**Funding:** The authors declare no sources of funding or other support for this study.

**Competing interests:** The authors certify that there are no conflicts of interests.

## Conclusion

Clinical profiles of post-COVID syndrome differed between groups. Muscle parameters were lower in the ICU group but patients who had not needed ICU admission had worse anxiety and HRQoL scores. Many patients who had not required mechanical ventilation had respiratory muscle weakness.

## Trial registration

ClinicalTrials.gov Identifier: NCT04852718

## Introduction

The persistence of symptoms after COVID infection is commonly referred to as post-acute COVID syndrome (PACS). Although there is no consensus, PACS is generally considered to include two subgroups, patients with prolonged or persistent PACS and patients with sequelae. Prolonged PACS syndromes are defined as the persistence of symptoms 4 weeks after the acute COVID infection, with an on-going, relapsing/remitting, or progressively improving course. The sequelae subgroup is defined as the presence of irreversible tissue damage 12 weeks after the acute disease. Such injuries can trigger varying degrees of permanent dysfunction and the corresponding symptomatology [1]. Mumoli et al. [2] suggest that these symptoms could be due to an aberrant immune response. The prevalence and clinical presentation of PACS is heterogeneous. The most frequent symptoms are fatigue, (52%), cardiorespiratory symptoms (mainly dyspnoea on exertion) (30–42%), and neurological symptoms (40%) [1].

According to various authors, following acute COVID-19 infection, approximately 45% of patients require healthcare support after discharge and around 5–10% have low functional capacity at 3, 6 and 12 months [3–6]. In a cohort of adult patients hospitalized for mild to severe COVID-19, Betschart et al. [3] found functional limitations persisted one year after hospitalization, and suggested that specific individualized support should be continued until full recovery. Along similar lines, another recent study [7] analyzed the need for health resources due to the persistence of symptoms in three profiles of post-COVID patients who presented sequelae at 6 months after COVID-19 infection (non-hospitalized, hospitalized, and ICU patients). The results showed that although the three groups of patients presented a high health burden, the more severe the acute illness, the greater the needs. The authors suggested that long-term multidisciplinary care is warranted for patients with sequelae of COVID-19.

Several guidelines have proposed specific and multidimensional rehabilitation programs to address this clinical situation [8, 9]. At our hospital, a post-COVID rehabilitation facility (MPCR) was established with the collaboration of a multidisciplinary team of rehabilitation physicians, pneumologists, physiotherapists, occupational therapists and speech therapists.

With the hypothesis that symptoms and limitations of post-COVID syndrome could differ according to the severity of the acute disease, our objective *was* to compare the patterns of symptoms in patients seen at our post-COVID rehabilitation unit in relation to the severity of acute COVID-19.

## Material and method

### Design

We performed a retrospective observational study based on medical records from the first consultation at the MPCR.

Inclusion criteria were: 1) patients who had had COVID-19 infection confirmed by a serological or molecular test (PCR); 2) living at home; 3) a negative PCR or IgM at initial evaluation in the MPCR or at least 28 days after diagnosis; 4) symptoms (mMRC ≥2 and/or clinical frailty scale value ≥3). Exclusion criteria were: 1) active COVID infection; 2) previous cognitive impairment; and 3) residence outside the hospital's health care service area.

The study was approved by the local clinical research ethics committee (IIBSP-COV-2020-155). Data collected were recorded expressly for the purpose of the study. Informed consent from patients was waived in view of the retrospective nature of the study. The trial was registered in ClinicalTrials.gov (NCT04852718).

## Evaluation measures

Our post-COVID rehabilitation consultation was started in June 2020. The patients were referred to our rehabilitation unit from primary care (GP), from the pneumology outpatient department, or at discharge from hospital.

The time elapsed between recovery from the acute disease and visit to the MPCR varied from 3 to 9 months.

From the medical records gathered at first visit to the MPCR we assessed the following variables: 1) the Barthel index (BI) [10]; 2) cognitive status according to the Pfeiffer scale [11]; 3) dyspnoea in daily activities according to the Medical Research Council (mMRC) scale [12]; 4) anxiety and depression with the HADS scale [13]; 5) health-related-quality-of-life (HRQoL) with the generic SF-36 questionnaire [14]; 6) muscle strength scale (mMRC) [15]; 7) the short physical performance battery (SPPB) for frailty [16]; 8) manual dynamometry using the Jamar Hydraulic Hand Dynamometer [17]; 9) effort capacity with the 6-minute walk test-6MWT [18];10) oxygen saturation ($SpO_2$) during the 6MWT; 11) respiratory muscle strength measured by maximum inspiratory pressure ($P_{Imax}$) and maximum expiratory pressure ($P_{Emax}$) [19]; with the Pocket Spiro MPM-100 device; and 12) the EAT-10 Dysphagia questionnaire [20].

Based on the results from this assessment, the team decided whether or not the patient's required rehabilitation.

## Statistical analysis

Analysis of variance (ANOVA) was used for quantitative variables, providing the mean and its corresponding standard deviation for each group. If significant, the post-hoc Scheffe test was also used. In the case of categorical variables, the Chi square was used, providing the absolute value (frequency) and its percentage. The Pearson correlation coefficient was used for quantitative variables, and the non-parametric Spearman test was used to validate the results.

The probability of a type I error was set to the value of 5% (alpha = 0.05). All analyses were performed using the statistical package IBM-SPSS statistics (V26.0).

## Results

### Sample description

From June 2020 to June 2021, we recorded data from 178 patients. There was a predominance of men (58.9% vs 41.1%), and mean age was 59.6±11.6 years. One hundred and fifty-two patients had required hospital admission, 91 of whom were admitted to the intensive care unit (ICU). The mean hospital stay for the total group of patients was 34.8±32.8 days (range from 1 to 227). Most of the patients attended had COVID-19 in the first wave of the disease. Regarding the patients admitted to the ICU or the conventional ward, the mean time from hospital

**Table 1. General characteristics of patients according to the three pandemic waves analysed.**

| PATIENTS | TOTAL | P | 1st Wave | P1 | 2nd Wave | P2 | 3rd Wave | P3 |
|---|---|---|---|---|---|---|---|---|
| **N (%)** | 178 | | 115 (64,6%) | | 19 (10.7%) | | 44 (24.7%) | |
| **Male n (%)** | 105 (58.9%) | **0.023** | 57 (49,7.4) | 0.055 | 14 (73.7%) | 0.794 | 31 (70.5%) | **0.021** |
| **Age (years)** | 59.6±11.6 | **0.034** | 58.9±11.9 | 0.494 | 55.3±13.6 | **0.046** | 63.0±8.8 | 0.139 |
| **ICU Admission (n/%)** | 91 (51.1%) | **< 0.001** | 43 (37.4%) | 0.117 | 10 (52.6%) | **0.014** | 38 (86.4%) | **< 0.001** |
| **Ward (n/%)** | 60(34%) | | 51 (44.3%) | | 4 (21.1%) | | 4 (9.1%) | |
| **No Admission (n/%)** | 27 (15.3%) | | 20(17.4%) | | 5 (26.3%) | | 2 (4.5%) | |
| **Hospital discharge / Visit (days)** | 109.7±79.6 | **< 0.001** | 136.7 ±87.5 | 0.221 | 100.3±55.4 | 0.141 | 56.4±20.5 | **< 0.001** |

**1st Wave** from March to May 2020; **2nd Wave** from July to October 2020; **3rd Wave**: from December 2020 to February 2021; p = comparison between the three waves; p1 = comparison between 1st wave and 2nd wave; p2 = comparison between 2nd and 3rd wave; p3 = comparison between 1st and 3rd wave.

discharge and the first visit to the MPCR was 109.7±79.6 days (range 12 to 432 days), this period was longest in patients admitted during the first wave than the third wave (Table 1).

Patients admitted to the ICU were older than patients in the other two groups and there were more males (p<0.001) (Table 1). ICU patients had a mean stay of 26.8±19.4 days (range 3 to 107). They all required invasive mechanical ventilation (IMV). Tracheotomy was needed in 34 (37.4%) patients, pronation techniques in 41 (45.1%) and ECMO in 10 (9.1%).

One hundred and fifty-two of the 178 patients (85.4%) had at least one comorbidity before COVID-19 infection without differences between groups (Table 2). These comorbidities were obesity (43.6%), arterial hypertension (AHT) (36.3%), respiratory disease (33%), and heart disease (13.4%). ICU Patients had a higher number of cardiovascular comorbidities such as heart disease (p = 0.02) and AHT (p = 0.01) than the other two groups.

The most frequent complications in patients admitted to the hospital (ICU or conventional ward) were muscular weakness (42.4%), neurological (19.2%), respiratory (18.1%), or cardiologic (9.6%) complications and pulmonary thromboembolism (8.5%). Thirteen patients of the 41 ICU (31.7%) patients who required pronation techniques had peripheral neurological complications (neuropathy of the tibial, axillary or ulnar nerves). There was a positive correlation (p<0.01) between these complications and pronation.

The symptoms reported at the first visit of the MPCR were fatigue (78.2%), dyspnoea (75.4%), effort intolerance (27%), neurological alterations (distal paraesthesia or motor deficit) (26%), and anxious-depressive syndrome 14.5%.

## Outcomes

The mean BI for all groups was 96.5±7.7 (range 64–100), with moderate dependence (BI<60) in 2% and mild dependence (BI<90) in 14.5% of patients. The BI was lower in ICU patients (p = 0.002).

Eleven patients (6.2%) showed cognitive impairment according to the Pfeiffer scale. These patients were all referred to the neuropsychology clinic and mild-moderate impairment was confirmed in 8. There were no differences between groups. In 25 patients (14%) who reported memory loss or difficulty concentrating, the Pfeiffer score was normal. In 14 of these 25 patients, a neuropsychological evaluation was done and 12 patients showed mild-moderate cognitive impairment. The total number of patients with cognitive impairment was therefore 20 (11.2%).

The mMRC scale revealed dyspnoea in 134 patients (75.3%), with a score of 1 in 56 and scores of 2 to 3 in the remaining group. There were no significant differences between groups.

**Table 2. Characteristics of the patients and results of the evaluation measures according to the severity profile.**

|  | TOTAL | p | ICU | P1 | WARD | P2 | No Admission | P3 |
|---|---|---|---|---|---|---|---|---|
| N | 178 |  | 91 (51.1%) |  | 60 (33.7%) |  | 27 (15.2%) |  |
| Male (n / %) | 105 (58.9%) | <0.001 | 65 (71.4%) | 0.008 | 30 (50%) | 0.145 | 9 (33.3%) | <0.001 |
| Age | 59.6±11.6 | <0.001 | 63.3 ± 9.9 | 0.025 | 58.5 ± 11.7 | 0.010 | 50.9±12.5 | < 0.001 |
| Comorbidities | 152 (85,4%) | 0.609 | 80 (87.9%) | 0.430 | 50 (83.3%) | 0.833 | 22 (81.5%) | 0.406 |
| Barthel Index | 96.5±7.7 | 0.002 | 94.5±9.6 | 0.012 | 98.2±4.5 | 0.896 | 99.3±3.8 | 0.014 |
| Cognitive impairment (Pfeiffer) n/% | 11 (6,2%) | 0.121 | 7 (7.8%) | 0.078 | 1 (1.7%) | 0.064 | 3 (11.1%) | 0.597 |
| mMRC dyspnoea (n /%) | 134 (75.3%) | 0.682 | 66 (72.5%) | 0.418 | 47 (78.3%) | 0.954 | 21(77.8%) | 0.581 |
| HAD anxiety n/% | 56 (31.5%) | 0.001 | 17 (18.7%) | <0.001 | 28 (46.7%) | 0.606 | 11(40.7%) | 0.023 |
| HAD depression n/% | 42 (23.6%) | 0.096 | 16 (17.6%) | 0.185 | 16 (26.7%) | 0.333 | 10 (37%) | 0.04 |
| SF36 PCS | 36.4±9.4 | 0.254 | 36.9±9.5 | 0.99 | 37±9.1 | 0.3 | 33.4±9.6 | 0.308 |
| SF 36 MCS | 44.6±11.8 | 0.007 | 47.7±10.9 | 0.008 | 41±11.1 | 0.658 | 43.6±13.8 | 0.397 |
| mMRC strength | 57.6±3.9 | <0.001 | 56.3±4.5 | <0.001 | 58.8±2.5 | 0.891 | 59.3±1.1 | 0.002 |
| SPPB | 9.6±2.6 | 0.155 | 9.3±2.8 | 0.269 | 10±2.3 | 0.954 | 10.2±2.1 | 0.335 |
| Dynamometry impairment n/% | 146 (82.5%) | 0.001 | 83 (92.2%) | <0.001 | 41 (68%) | 0.193 | 22 (81.5%) | 0.129 |
| P6MM (meters) | 435.9±138 | 0.001 | 403.8±140.8 | 0.120 | 450.1±138.9 | 0.145 | 512.1±85.2 | 0.002 |
| Desaturation (SpO$_2$) (n) | 26 (14.6%) | 0.006 | 18 (69.2%) | 0.298 | 8 (30.8%) | 0.012 | 0 | 0.001 |
| PImax (cmH$_2$O) | 79.1±26.6 | 0.279 | 82.1±28.2 | 0.51 | 76.9±27.1 | 0.89 | 73.9±18.1 | 0.38 |
| PEmax (cm H$_2$O) | 101.8±27.3 | 0.881 | 102.5±24.3 | 0.9 | 100.3±34.9 | 0.93 | 102.7±16.1 | 0.99 |

ICU: patients admitted to the intensive care unit, general ward, and not admitted. mMRC: Modified Medical Research Council scale; HAD: hospital anxiety and depression questionnaire SF36: quality of life questionnaire; 6MWT: 6-minute walk test; SpO2: oxyhemoglobin saturation; PImax: maximum inspiratory pressure; PEmax: maximum expiratory pressure. p = comparison between the three groups; p1 = comparison between ICU-ward; p2 = comparison between ward—no admission; p3 = comparison between ICU-no admission.

According to the HADs scale, anxiety was detected in 56 patients, with a mean value of 6.27 ±4.2 (range 1 to 21). Anxiety was more common in patients admitted to a general ward and in those not admitted to hospital than in patients admitted to the ICU (p<0.001). Depression was diagnosed in 42 patients, with a mean value of 5.27±3.8 (range 1 to 15). It was more common in patients who were not admitted to hospital (p = 0.04).

HRQoL showed a clear deterioration with a mean value of 36.4±9.4 in the physical component and 44.6±11.8 in the mental component of the SF36 questionnaire. ICU patients had a better score in the mental component than those in the ward (p = 0.008).

The peripheral muscle strength study showed a mean value of the mMRC scale of 57.6 ±3.9 (38–60), being significantly lower in ICU patients (p<0.001).

The mean value of manual dynamometry was 16.6±7.2. We found values lower than reference values in 146 patients; 92.2% in the ICU, 68% in patients admitted to the ward, and 81.5% in not admitted patients (p<0.001).

The mean SPPB value was 9.6±2.6, without differences between groups. The value was below 9 in 61 patients (34.1%). Six of 61 patients (9.8%) were classified as disabled, 19 (31.1%) as frail and 36 (59%) as pre-frail.

The mean distance on the 6MWT was 435.9±138 (range 75 to 750) meters. Only eight patients had scores below the reference value. Patients admitted to the ICU walked less distance than those who were not admitted (p = 0.002), but there were no differences with those admitted to the ward.

Twenty-six patients presented desaturation (mean SpO$_2$ < 90%) during the 6MWT, 18 in ICU patients, 8 in patients admitted to the ward, and none in the not admitted group. CO

diffusion capacity (DLco) was only available for 15 of the 26 patients. Of these, 8 presented a decrease and 7 were normal.

The mean value of $P_{Imax}$ was 79.1±26.6 $cmH_2O$ (range 23 to 157). In 57 patients (31.8%) the value was below the reference value. Twenty-seven of the 57 were admitted to the ICU and required IMV. The mean $P_{Emax}$ value, was 101.8±27.3 $cmH_2O$, within the reference range in the three groups.

Swallowing disorders were detected in 16 patients: 8 ICU patients who required IMV, 7 general ward patients who did not require IMV, and 1 patient who did not need hospital admission.

The time elapsed between hospital discharge and first visit to the MPCR showed a positive correlation with BI, mMRC force and 6MWT score (p<0.001). We also found a positive correlation between the value of manual dynamometry, mMRC force, 6MWT and the physical component of the SF36 questionnaire (p = 0.001).

See all results in Table 2.

## Discussion

In this comparison of post-COVID syndrome according to the severity of the acute disease, ICU patients had more severe muscular weakness, less anxiety and better HRQoL scores than patients in the conventional ward and patients not admitted to hospital.

In reference to BI, our patients showed a mild level of dependence. This contrasts with findings from other studies. Curci et al. [21] and Belli et al. [22] found a high level of dependence (BI<60) in patients admitted to their rehabilitation department, without differences between those who needed IMV and those who did not. The difference between our results and those of Belli and Curci may be because they evaluated patients immediately after hospital discharge while we evaluated patients at least 3 months after discharge. We found a positive correlation between the time from discharge and degree of functional capacity.

The Pfeiffer questionnaire showed low ability to detect cognitive impairment in our patients. This could explain the differences with the results from Johnsen et al. [23]. At the three-month follow up, these authors observed cognitive impairment in over 50% of their patients, predominantly in those who were hospitalized. In contrast, in our series we only found cognitive impairment in 11,2% of patients in the same clinical conditions as in Johnsen et al. study.

Dyspnoea appears to be a prevalent symptom in all studies [4–6, 24] and our findings coincide with these previous observations. The mMRC scale showed a good ability to detect dyspnoea in all three groups.

Also, in accordance with previous studies, we found anxiety and depression were high in patients with post-COVID syndrome. Several authors [4–6, 24–26] has attributed this to impaired functional capacity. Wang et al. [26] suggest male gender, older age, and less use of social media as predisposing factors. It is of note in our sample that patients admitted to the UCI had lower anxiety. We hypothesize that this may be due to the intensive monitoring since admission, the prompt rehabilitation, and patients' relief at having overcome a critical situation.

Similarly, to other authors who measured HRQoL in post COVID patients we observed low values in all patients despite the different questionnaires used [27–30]. HRQoL is considered to be influenced by factors such as age, the need or not for IMV, length of hospital stay, male gender, comorbidities, and persistence of symptoms of the infection itself [28, 29]. In our sample the best scores in the SF-36 mental component were observed in ICU patients. Again, we hypothesize that the close supervision and earlier rehabilitation play an important role.

Muscular weakness is a predominate clinical sign in all series published to date [5, 6, 21, 22, 24] but few specify which tests were performed. Using SPPB, as in our study, Belli et al. [22], found functional capacity was worse than in our study, likely because they did perform this evaluation immediately after hospital discharge. However, like these authors, we not find differences between patients who needed IMV and those who did not. Therefore, coinciding with findings by Tanriverdi et al. [31], we found that ICU patients showed lower strength values than the other two groups according to measurements observed for mMRC force and manual dynamometry.

Concerning the 6MWT, although ICU patients covered a shorter distance, almost all our participants had a score within the reference value. These results support those of Eksombatchai et al. [32] who found no significant differences between patients with severe or mild disease at 2 months after COVID. However, our findings contrast with those of other authors who reported patients did not manage to finish the test or to walk less than 200 meters a several weeks [21] or even at 3 months after hospital discharge [27]. The high proportion of effort desaturation in our series contrast with that found by other authors [27, 32]. The reason for this difference could be because they did not measure this parameter during the 6MWT.

Respiratory muscles weakness is a frequent finding in COVID-19 patients receiving IMV. Using ultrasound, Farr et al. [33] showed those patients with COVID-19 and IMV treatment had a greater reduction in the contractile capacity of the diaphragm than patients with the same characteristics but without COVID. Moreover, Abodonya et al. [34] suggested that $P_{Imax}$ could be altered in patients with post-Covid syndrome. Although they did not measure $P_{Imax}$, they showed that respiratory muscle training improves symptoms, lung function, HRQoL, and exercise capacity. In our study, we observed a high percentage of patients had a decrease in $P_{imax}$ and more than half of them did not require IMV. We hypothesize that inspiratory muscle weakness maybe is due to factors other than IMV, such as the lack of early physiotherapy in non-ICU patient's or the infection itself.

## Strengths and limitations of the study

The main limitations of this study are its retrospective design and the different time lapses between recovery from the acute disease and performance of the MPCR. Moreover, the convenience sampling applied caused an imbalance in the distribution of the three groups.

Finally, fatigue was assessed only from the symptoms reported by the patients.

The main strength of our study is the comparison of post-COVID symptoms in three severity profiles of acute infection. This has been previously studied by Ziyed et al. [7] but our study included two further important aspects, HRQoL and respiratory muscle strength.

## Conclusions

On comparing the clinical characteristics of post-COVID-19 in three severity profiles, we found muscle parameters were lower in the ICU group, anxiety and HRQoL scores were worse in patients who had not needed ICU admission, and respiratory muscle weakness was worse in patients who had not required mechanical ventilation.

We consider these findings emphasize the relevance of early rehabilitation and assessment of respiratory muscle strength in patients with post-COVID syndrome.

## Supporting information

**S1 Data.**
(XLSX)

## Acknowledgments

The authors would like to thank to Dr Carme Puy for her support in the study, and the physiotherapists, occupational and speech therapists for their participation.

## Author Contributions

**Conceptualization:** Jean Claude Perrot, Macarena Segura, Marta Beranuy, Mª Rosa Güell-Rous.

**Data curation:** Jean Claude Perrot, Ignasi Gich, Mª Rosa Güell-Rous.

**Formal analysis:** Ignasi Gich.

**Investigation:** Macarena Segura, Marta Beranuy, Mª Josepa Nadal, Alberto Pintor, Jimena Terra, Eliot Ramirez, Luis Daniel Paz, Mª Rosa Güell-Rous.

**Methodology:** Jean Claude Perrot, Macarena Segura, Marta Beranuy, Mª Rosa Güell-Rous.

**Supervision:** Helena Bascuñana, Vicente Plaza.

**Writing – original draft:** Jean Claude Perrot, Mª Rosa Güell-Rous.

**Writing – review & editing:** Jean Claude Perrot, Mª Rosa Güell-Rous.

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
