## [Decision Letter · Decision Letter 0]

31 May 2022

PONE-D-22-11582“ Comparison of post-COVID symptoms in patients with different severity profiles of the acute disease”PLOS ONE

Dear Dr. Rous,

Thank you for submitting your manuscript to PLOS ONE. After careful consideration, we feel that it has merit but does not fully meet PLOS ONE’s publication criteria as it currently stands. Therefore, we invite you to submit a revised version of the manuscript that addresses the points raised during the review process.

We look forward to receiving your revised manuscript.

Kind regards,

Peter Schwenkreis

Academic Editor

PLOS ONE

Journal Requirements:

2. Please provide additional details regarding participant consent. In the Methods section, please ensure that you have specified (1) whether consent was informed and (2) what type you obtained (for instance, written or verbal). If your study included minors, state whether you obtained consent from parents or guardians. If the need for consent was waived by the ethics committee, please include this information.

Reviewers' comments:

Reviewer's Responses to Questions

**Comments to the Author**

1. Is the manuscript technically sound, and do the data support the conclusions?

Reviewer #1: Partly

Reviewer #2: Partly

2. Has the statistical analysis been performed appropriately and rigorously? 

Reviewer #1: Yes

Reviewer #2: I Don't Know

3. Have the authors made all data underlying the findings in their manuscript fully available?

Reviewer #1: No

Reviewer #2: Yes

4. Is the manuscript presented in an intelligible fashion and written in standard English?

Reviewer #1: Yes

Reviewer #2: Yes

5. Review Comments to the Author

Reviewer #1: The authors describe symptoms of patients who attended a Post-COVID Rehabilitation Consultation after acute COVID-19. Patients are categorized in no hospital treatment, hospital treatment, ICU and ventilation treatment. However, the baseline or reference groups for these categories are not known. I think the paper is interesting, however some weaknesses should be discussed. I hope my comment will be helpful

Comments

1. Abstract: First sentence

Little is known about the impact of the severity of COVID-19 on post-COVID sequelae.

I think this is not a fair statement. I would think, it is appropriate to write: Following literature, severity of COVID-19 seems to impact on post-COVID sequelae.

2. Introduction: Is the following sentence correct? The sequelae subgroup is defined as the presence of irreversible tissue damage 12 weeks …… Is it tissue damage or is it the persistence or emergence of symptoms?

3. I think you will have to elaborate on your study question. Now you write: “With the hypothesis that symptoms and limitations of post-COVID syndrome could differ according the severity of the acute disease, our objective was to compare these data in three severity profiles of COVID-19 infection.” However, all patients enrolled in your study took part in the Post-COVID Rehabilitation Consultation. Therefore, it is not possible to study the influence of severity of acute COVID-19 on Post-COVID symptoms. What you can study is the different patterns of symptoms in patients admitted to a rehabilitation program depending on severity of acute COVID-19. I think this is a relevant study questions as the patients might have different needs depending on symptoms.

4. The previous point considered, I suggest to think about the title of your paper. Isn’t it more a descriptive study of symptoms shown by patients coming for Post-COVID consultation to a clinic?

5. The introduction is very short. I think at least a few studies on risk factors for Post-COVID should be mentioned. Stick to the larger ones or the more recent ones for not getting overwhelmed.

6. Did you calculate a response rate?

7. Discussion: you write: “The Pfeiffer questionnaire showed low sensitivity to detect cognitive impairment in our patients.” Is it really sensitivity? Do you know the real proportion of patients with cognitive impairment?

8. Limitations and strength: your write: The main strength is that, to our knowledge, it is the first study to compare post-COVID symptoms in three severity profiles of acute infection. I am afraid, this is not correct. For an example please see Ziyad Al-Aly, Yan Xie, Benjamin Bowe Nature | Vol 594 | 10 June 2021.

8. Limitations: It should be mentioned that the non-hospital group is very small.

Good luck with the revision

Reviewer #2: My major comment concerns the different group seizes. I am no statistic expert, the very different group seizes complicate to rate the value of the results.

I have a view other comments:

Introduction: The sequelae subgroup is defined as the presence of irreversible tissue damage 12 weeks after the acute disease.

This is a very strict and clinically not helpful definition. How can a clinician know that the presenting symptoms are due to irreversible tissue damage? In particular the symptom fatigue eludes the problems of this definition.

It maybe helpful to make in the abstract transparent, that only patients with persistent dyspnoea or fraility were included in the study.

The assessment battery is differentiated. How were fatigue and fatigability measured?

How did the patients find the rehabilitation department? Was there a kind of admission by their GP? Could patients present themselves?

In the discussion section the authors stated, that dyspnoea appears to be a prevalent symptom, but this was one of their inclusion criteria.

History is a good predictor of anxiety and depression. How many patients had a history of anxiety and depression?

Quality of life is influenced by the presence of anxiety and depression. Not surprisingly the group with the lowest incidence of anxiety and depression has the best quality of life.

Concerning myopathy, is ICU acquired weakness, critical illness polyneuropathy and critical illness myopathy in the focus of the authors? How was the diagnosis myopathy made? Were there clinical neurophysiologic examinations? Was the diagnosis based on clinical examination? How likely were the acquired weaknesses after ICU stay due to a COVID specific pathophysiologic reason?

Concerning the conclusion, the effects of early rehabilitation were not the topic of the paper.

Why is the assessment of respiratory muscle strength important?

Why are anxiety and HRQol worse in non ICU patients? This two items may have a correlation. It could make sense to treat anxiety and depression.

6. PLOS authors have the option to publish the peer review history of their article (what does this mean?). If published, this will include your full peer review and any attached files.

Reviewer #1: **Yes: **Albert Nienhaus

Reviewer #2: No

---

## [Author Response · Author response to Decision Letter 0]

29 Jun 2022

REVIEW COMMENTS TO THE AUTHOR

Peter Schwenkreis, Academic Editor

The manuscript meets the requirements of PlosOne journal, in our opinion.

2. Please provide additional details regarding participant consent. 

The patients did not complete the informed consent, since it is a retrospective study, based on the data obtained in the medical records collected in our rehabilitation clinic.

We have added the following sentence in Material and methods:

“Informed consent from patients was waived in view of the retrospective nature of the study”. 

3. PLOS requires an ORCID iD for the corresponding author in Editorial Manager on papers submitted after December 6th, 2016. 

The corresponding author has the ORCID ID

 

Reviewer #1: 

The authors describe symptoms of patients who attended a Post-COVID Rehabilitation Consultation after acute COVID-19. Patients are categorized in no hospital treatment, hospital treatment, ICU and ventilation treatment. However, the baseline or reference groups for these categories are not known. I think the paper is interesting, however some weaknesses should be discussed. I hope my comment will be helpful

Comments

1. Abstract: First sentence

Little is known about the impact of the severity of COVID-19 on post-COVID sequelae.

I think this is not a fair statement. I would think it is appropriate to write: Following literature, severity of COVID-19 seems to impact on post-COVID sequelae.

We have now reworded this sentence in the revised manuscript to read… 

“Studies in the literature suggest the severity of COVID-19 may impact on post-COVID sequelae”

2. Introduction: Is the following sentence, correct? The sequelae subgroup is defined as the presence of irreversible tissue damage at 12 weeks. 

Yes, this is correct. We defined the sequelae in accordance with the definition from a consensus in Catalonia (reference nº 1)

Is it tissue damage or is it the persistence or emergence of symptoms?

Regarding tissue damage or persistence of symptoms, this is not yet known. This point was discussed in the editorial of Munoli (reference 2) and in our consensus (reference 1). Because it is a very new clinical entity, we probably need more time to understand this.

3. I think you will have to elaborate on your study question. Now you write: “With the hypothesis that symptoms and limitations of post-COVID syndrome could differ according the severity of the acute disease, our objective was to compare these data in three severity profiles of COVID-19 infection.” However, all patients enrolled in your study took part in the Post-COVID Rehabilitation Consultation. Therefore, it is not possible to study the influence of severity of acute COVID-19 on Post-COVID symptoms. What you can study is the different patterns of symptoms in patients admitted to a rehabilitation program depending on severity of acute COVID-19. I think this is a relevant study questions as the patients might have different needs depending on symptoms.

Thank you for this suggestion.

We have now rewritten the objective (in the Abstract and in the Introduction), adding that the study is about patients visited at our rehabilitation unit. 

In the abstract: 

“We retrospectively compared the different patterns of symptoms in relation to the severity of acute COVID-19in patients visited at our post-COVID rehabilitation unit.” 

In the introduction: “Our objective was to compare the patterns of symptoms in patients seen at our post-COVID rehabilitation unit in relation to the severity of acute COVID-19. “

4. The previous point considered; I suggest to think about the title of your paper. Isn’t it more a descriptive study of symptoms shown by patients coming for Post-COVID consultation to a clinic?

Yes, thank you. We have now added “visited at a rehabilitation unit” 

“Comparison of post-COVID symptoms in patients with different severity profiles of the acute disease visited at a rehabilitation unit”

5. The introduction is very short. I think at least a few studies on risk factors for Post-COVID should be mentioned. Stick to the larger ones or the more recent ones for not getting overwhelmed.

We have now included findings from other studies as suggested by the reviewer: The text now reads: 

“According to various authors, following acute COVID-19 infection, approximately 45% of patients require healthcare support after discharge and around 5-10% have low functional capacity at 3, 6 and 12 months3-6. In a cohort of adult patients hospitalized for mild to severe COVID-19, Betschart et al3 found functional limitations persisted one year after hospitalization, and suggested that specific individualized support should be continued until full recovery. Along similar lines, another recent study7 analyzed the need for health resources due to the persistence of symptoms in three profiles of post-COVID patients who presented sequelae at 6 months after COVID-19 infection (non-hospitalized, hospitalized, and ICU patients). The results showed that although the three groups of patients presented a high health burden, the more severe the acute illness, the greater the needs. The authors suggested that long-term multidisciplinary care is warranted for patients with sequelae of COVID-19.”

6. Did you calculate a response rate? 

We could not calculate a sample rate because we analyzed the results retrospectively. All the patients were included.

7. Discussion: you write: “The Pfeiffer questionnaire showed low sensitivity to detect cognitive impairment in our patients.” Is it really sensitivity? Do you know the real proportion of patients with cognitive impairment?

In our opinion, the Pfeiffer questionnaire was not sensitive in our patients. As this is a descriptive study only, we cannot determine the exact proportion of patients with this impairment.

8. Limitations and strength: your write: The main strength is that, to our knowledge, it is the first study to compare post-COVID symptoms in three severity profiles of acute infection. I am afraid, this is not correct. For an example please see Ziyad Al-Aly, Yan Xie, Benjamin Bowe Nature | Vol 594 | 10 June 2021.

We thank the reviewer for the bibliographic citation of Ziyed et al. We were unaware of this study. It is indeed important to consider it because it compares the same patient profiles as in our study. We have now added a paragraph in the introduction (see point 5) and added a comment in the section of limitations and strengths. It reads as follows:

“The main strength of our study is the comparison of post-COVID symptoms in three severity profiles of acute infection. This has been previously studied by Ziyed et al7 but our study included two further important aspects, HRQoL and respiratory muscle strength.”

.

9. Limitations: It should be mentioned that the non-hospital group is very small. 

We think we have made this clear now too. 

“The main limitations of this study are its retrospective design and the different time lapses between recovery from the acute disease and performance of the MPCR. Moreover, the convenience sampling applied caused an imbalance in the distribution of the three groups” 

Good luck with the revision. Thank you very much for the helpful suggestions.

 

Reviewer #2: 

My major comment concerns the different group seizes. I am no statistic expert; the very different group seizes complicate to rate the value of the results.

As our study is descriptive and with a convenience sampling, there is an imbalance in the distribution of the three groups of patients, as expected, according to the severity of the disease.

We have now added this limitation in the manuscript…” 

Moreover, the convenience sampling applied caused an imbalance in the distribution of the three groups.”

1.- Introduction: The sequelae subgroup is defined as the presence of irreversible tissue damage 12 weeks after the acute disease.

This is a very strict and clinically not helpful definition. How can a clinician know that the presenting symptoms are due to irreversible tissue damage? In particular, the symptom fatigue eludes the problems of this definition.

1a). The sequelae subgroup is defined as the presence of irreversible tissue damage 12 weeks after the acute disease. This is a very strict and clinically not helpful definition.

This is correct. We cannot know if the symptoms are due to irreversible tissue damage. However, our comment is based on the clinical definition from the consensus in our country (reference nº 1) :

1. Long-COVID: persistence of symptoms (present or not at the onset of the infection) after 4 weeks of infection, with a permanent, relapsing / remitting or progressive improvement course. 

2. Sequelae: irreversible tissue damage after 12 weeks that could trigger different degrees of permanent dysfunction and associated symptomatology. 

How can a clinician know that the presenting symptoms are due to irreversible tissue damage?

Regarding tissue damage or persistence of symptoms, this is not yet known. This point was discussed in the editorial of Munoli (reference 2) and also in our consensus (reference 1). Because it is a very new clinical entity, we probably need more time to understand this.

2. - It maybe helpful to make in the abstract transparent, that only patients with persistent dyspnoea or fraility were included in the study.

Thank you. We have now added these two selection criteria in the abstract:

 “The main inclusion criteria were persistent dyspnoea (mMRC ≥2) and/or clinical frailty (scale value ≥3).”

3.- The assessment battery is differentiated. How were fatigue and fatigability measured?

Fatigue is a symptom, so, we based this on what the patients told us. We did not use any specific scale. However, it was related to the other outcomes, such as the Borg scale during the 6MWT, muscle strength measures (SPPB, mMRC muscle strength scale) and even the frailty scale. We did not analyse fatigability in our study.

 4- How did the patients find the rehabilitation department? Was there a kind of admission by their GP? Could patients present themselves?

We have now clarified this in the Material and Methods:

“The patients were referred to our rehabilitation unit from primary care (GP), from the pneumology outpatient department, or at discharge from hospital”. 

5.- In the discussion section the authors stated, that dyspnoea appears to be a prevalent symptom, but this was one of their inclusion criteria.

We stated that the main inclusion criteria were “persistent dyspnoea (mMRC ≥2) and/or clinical frailty (scale value ≥3)”. Over 75% of patients had dyspnoea.

6.- History is a good predictor of anxiety and depression. How many patients had a history of anxiety and depression?

We did not analyze this point. We reviewed the charts and we found scarce data about previous history of anxiety and depression. 

The only comorbidities we found were arterial hypertension, respiratory disease, and heart disease.

7.- Quality of life is influenced by the presence of anxiety and depression. Not surprisingly the group with the lowest incidence of anxiety and depression has the best quality of life.

Indeed, yes, we agree they are closely correlated.

8.- Concerning myopathy, is ICU acquired weakness, critical illness polyneuropathy and critical illness myopathy in the focus of the authors? 

No, ICU acquired weakness, critical illness polyneuropathy and critical illness myopathy was not our focus. 

How was the diagnosis myopathy made? 

The diagnosis of myopathy was make using: the muscle strength scale (mMRC); the short physical performance battery (SPPB), the manual dynamometry and the effort capacity with the 6-minute walk test-6MWT.

Were there clinical neurophysiologic examinations? No. 

Was the diagnosis based on clinical examination? Yes

How was the diagnosis myopathy made? It was a clinical diagnosis

Was the diagnosis based on clinical examination? Yes, using different scales

How likely were the acquired weaknesses after ICU stay due to a COVID specific pathophysiologic reason? 

As this was a clinical diagnosis we cannot demonstrate whether this myopathy was specific for COVID or for ICU stage. 

9.- Concerning the conclusion, the effects of early rehabilitation were not the topic of the paper.

 No, but this was written as a take-home message. 

“We consider these findings emphasize the relevance of early rehabilitation and assessment of respiratory muscle strength in patients with post-COVID syndrome”.

10.- Why is the assessment of respiratory muscle strength important?

The assessment of respiratory muscle is important because it could be a factor for effort dyspnea, as we discuss in the Discussion section.

In the follow-up patients, with weakness of respiratory muscles (PImax decreased) we start specific respiratory muscle training, and we found improvement in PImax and dyspnea. However, this finding will be a part of a second paper.

Why are anxiety and HRQol worse in non-ICU patients? 

We do not have a good explanation. We only can hypothesize, as we did in the discussion section.

These two items may have a correlation. 

Yes, we found a clear correlation between these two symptoms and the HRQoL higher than -0.60 with a p<0.001.

It could make sense to treat anxiety and depression.

We agree. We discuss this with patients and offer referral to the mental health service.

We thank the reviewers for all these helpful suggestions.

---

## [Editor Report · Decision Letter 1]

10 Jul 2022

PONE-D-22-11582R1“ Comparison of post-COVID symptoms in patients with different severity profiles of the acute disease visited at a rehabilitation unit ”PLOS ONE

Dear Dr. Rous,

Thank you for submitting your manuscript to PLOS ONE. After careful consideration, we feel that it has merit but does not fully meet PLOS ONE’s publication criteria as it currently stands. Therefore, we invite you to submit a revised version of the manuscript that addresses the points raised during the review process.

In my opinion, you have not sufficiently responded to 2 issues that were raised by the reviewers. The first issue is the use of the term "sensitivity" with respect to the Pfeiffer questionnaire. "Sensitivity" (true positive rate) refers to the probability of a positive test, conditioned on truly being positive. Since you do not know the real proportion of patients with cognitive impairment, you must not use the term "sensitivity" or "sensitive" in this context. The second issue is the use of the term "myopathy" in your study. Apparently you use this term synonymously to muscular weakness, based on the muscle strength scale (mMRC), the short physical performance battery (SPPB), the manual dynamometry and the effort capacity with the 6-minute walk test-6MWT. However, muscular weakness as assessed by these tests is only a symptom, whereas myopathy is defined as a disease of the muscles resulting e.g. from specific inflamatory or metabolic processes. Muscular weakness is not necessarily caused by myopathy, but may also occur in deconditioned patients due to immobilization. You should therefore avoid the use of the term "myopathy" in this context, when you cannot be sure that the patients are indeed suffering from a specific muscular disease. Besides, you should include in the limitations section that "fatigue" was only assessed by patients reportings, not by standardized and validated questionnaires.

We look forward to receiving your revised manuscript.

Kind regards,

Peter Schwenkreis

Academic Editor

PLOS ONE
---

## [Author Response · Author response to Decision Letter 1]

16 Aug 2022

RESPONSE TO THE ACADEMIC EDITOR, Peter Schwenkreis,

1.- The first issue is the use of the term "sensitivity" with respect to the Pfeiffer questionnaire. "Sensitivity" (true positive rate) refers to the probability of a positive test, conditioned on truly being positive. Since you do not know the real proportion of patients with cognitive impairment, you must not use the term "sensitivity" or "sensitive" in this context.

In agreement with the editor, we have changed the term "sensitivity" for the “ability”.

2.- The second issue is the use of the term "myopathy" in your study. Apparently you use this term synonymously to muscular weakness, based on the muscle strength scale (mMRC), the short physical performance battery (SPPB), the manual dynamometry and the effort capacity with the 6-minute walk test-6MWT. However, muscular weakness as assessed by these tests is only a symptom, whereas myopathy is defined as a disease of the muscles resulting e.g. from specific inflamatory or metabolic processes. Muscular weakness is not necessarily caused by myopathy, but may also occur in deconditioned patients due to immobilization. You should therefore avoid the use of the term "myopathy" in this context, when you cannot be sure that the patients are indeed suffering from a specific muscular disease.

Thank you for pointing this out. We agree and have now changed the term “myopathy” to “muscular weakness”.

3.- Besides, you should include in the limitations section that "fatigue" was only assessed by patients reportings, not by standardized and validated questionnaires.

In agreement with the editor, now we have added the following sentence in the limitations section:

Finally, fatigue was assessed only from the symptoms reported by the patients.

We thank the Editor for these helpful suggestions.

---

## [Editor Report · Decision Letter 2]

30 Aug 2022

“ Comparison of post-COVID symptoms in patients with different severity profiles of the acute disease visited at a rehabilitation unit ”

PONE-D-22-11582R2

Dear Dr. Rous,

We’re pleased to inform you that your manuscript has been judged scientifically suitable for publication and will be formally accepted for publication once it meets all outstanding technical requirements.

Kind regards,

Peter Schwenkreis

Academic Editor

PLOS ONE
---

## [Editor Report · Acceptance letter]

7 Sep 2022

PONE-D-22-11582R2 

Comparison of post-COVID symptoms in patients with different severity profiles of the acute disease visited at a rehabilitation unit 

Dear Dr. Güell-Rous:

I'm pleased to inform you that your manuscript has been deemed suitable for publication in PLOS ONE. Congratulations! Your manuscript is now with our production department. 

Kind regards, 

on behalf of

Dr. Peter Schwenkreis 

Academic Editor

PLOS ONE